# Neural Lattice Reduction:
# A Self-Supervised Geometric Deep Learning Approach

**Giovanni Luca Marchetti**[*]                                   *glma@kth.se*
*Royal Institute of Technology (KTH)*
*Stockholm*

**Gabriele Cesa, Pratik Kumar, Arash Behboodi**          *gcesa@qti.qualcomm.com*
*Qualcomm AI Research*[†]                                  *kpratik@qti.qualcomm.com*
*Amsterdam*                                                *behboodi@qti.qualcomm.com*

**Reviewed on OpenReview:** *https://openreview.net/forum?id=YxXyRSlZ4b*

## Abstract

Lattice reduction is a combinatorial optimization problem aimed at finding the most orthogonal basis in a given lattice. The Lenstra–Lenstra–Lovász (LLL) algorithm is the best algorithm in the literature for solving this problem. In light of recent research on algorithm discovery, in this work, we would like to answer this question: is it possible to parametrize the algorithm space for lattice reduction problem with neural networks and find an algorithm without supervised data? Our strategy is to use equivariant and invariant parametrizations and train in a self-supervised way. We design a deep neural model outputting factorized unimodular matrices and train it in a self-supervised manner by penalizing non-orthogonal lattice bases. We incorporate the symmetries of lattice reduction into the model by making it invariant to isometries and scaling of the ambient space and equivariant with respect to the hyperocrahedral group permuting and flipping the lattice basis elements. We show that this approach yields an algorithm with comparable complexity and performance to the LLL algorithm on a set of benchmarks. Additionally, motivated by certain applications for wireless communication, we extend our method to a convolutional architecture which performs joint reduction of spatially-correlated lattices arranged in a grid, thereby amortizing its cost over multiple lattices.

## 1 Introduction

Lattices are discrete geometric objects representing 'high-dimensional grids' that are ubiquitous in mathematics and computer science. In particular, two fundamental computational problems are based on lattices: the *shortest vector problem* (SVP) and the *closest vector problem* (CVP). These arise in several areas, among which asymmetric post-quantum cryptography (Hoffstein et al., 1998; Regev, 2009) and multi-input multi-output (MIMO) digital signal decoding (Hassibi & Vikalo, 2005; Worrall et al., 2022). Unfortunately, both are known to be computationally hard and no efficient algorithms exist to address them exactly.

The above-mentioned problems are easier to solve on lattices with highly-orthogonal bases. This has motivated the introduction of *lattice reduction* – another computational problem consisting of finding the most orthogonal basis for a given lattice. The problem is NP-hard, but can be solved approximately in polynomial time. The Lenstra–Lenstra–Lovász (LLL) algorithm (Lenstra et al., 1982) is a celebrated method for this purpose, and can be thought as a discrete analogue of the Gram-Schmidt orthogonalization procedure. LLL has been the subject of many theoretical and experimental studies, see (Smart et al., 2003; Nguyen & Vallée; Smart et al., 2003; Galbraith, 2012; Cohen, 2013) for more details.

---

[*]Work done during the internship in Qualcomm AI research.

[†]Qualcomm AI Research is an initiative of Qualcomm Technologies, Inc.

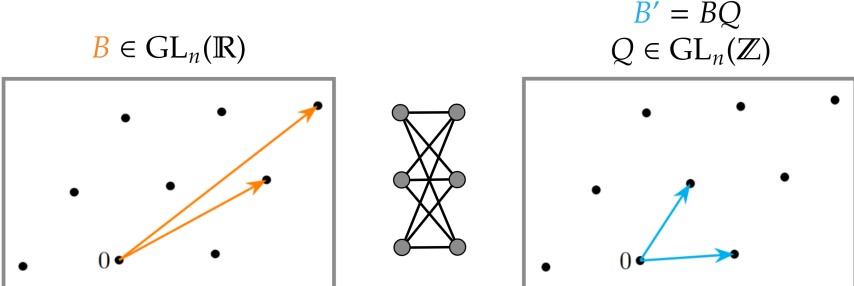

Figure 1: Our neural network maps a basis $B$ of a lattice to a reduced basis $B' = BQ$, where $Q$ is a unimodular base change.

However, in this work, we are adopting a different perspective. Considering the goal of lattice reduction algorithms is to construct a *more* orthogonal basis, can we find a new lattice reduction method by searching over a space that is parametrized by a class of neural networks?

A motivation is that data-driven approaches are able to discover or leverage complex patterns, potentially leading to better solutions (in terms of complexity or performance) for a specific distribution than those found by generic classical algorithms. In general, one can even try to use data to tune the hand-crafted parameters of a classical algorithm, which may result in sub-optimal solutions. Another motivation is the amortization capability of neural networks. When the lattice reduction needs to be applied jointly to multiple correlated lattices, neural networks with their parallel computing can help to amortize the overall computation cost.

A prominent example is the integer least square problem and its application in multiple-input multiple-output (MIMO) detection problem (Hassibi & Boyd, 1998; Hassibi & Vikalo, 2005), as lattice reduction is one of the main solutions to this problem. In digital communication, the messages are mapped to the points in a finite dimensional lattice, which are transmitted through the communication chain. Quadrature amplitude modulation (QAM) constellations are well known examples of such lattices, where $2^k$-QAM for $k$ up to 15 is used in various communication systems. With various techniques, the problem of recovering the transmitted messages boils down to finding the solution to a noisy linear inverse problem on the transmission lattice, which is an instance of integer least square problems. For multiple antenna systems, the transmission lattice is a multi-dimensional complex valued lattice, and the respective inverse problem is known as MIMO detection or demapping. In modern wireless protocols, e.g. 5G New Radio (NR), communication happens over a frequency-time grid (resource grid) where each grid point represents a communication channel for which the input lattice point needs to be recovered. Typically, this amounts to solving thousands of lattice reduction problems, thereby, incurring huge complexity cost. However, the channels for different grids are correlated thanks to the underlying physics of propagation: the correlation among adjacent lattices on the grid further motivates our deep learning approach which can leverage it to jointly reduce lattices on the grid at a reduced compute cost.

In this work, we develop a neural network model that outputs a base-change unimodular matrix given a lattice basis as input – see Figure 1. There are *two desiderata* for our approach. First, we will not leverage supervised learning for training the model. We are just given a set of matrices and the goal, which is to reduce the orthogonality metric. This is a natural choice, as the ground truth for the change of basis is not known for many cases. In this work, we minimize a measure of orthogonality for the output bases, and therefore the training objective is self-supervised.

Second, we would like to incorporate the symmetries of the problem into the model: this will provide hard guarantees about the generalization of the model under symmetric transformations. We show the limitations of including all the symmetries, particularly those related to the unimodular group. Nonetheless, we propose an architecture incorporating many symmetries of lattice reduction into the neural network. Specifically, we design a deep model that is simultaneously invariant and equivariant to the orthogonal group and to a finite subgroup of the unimodular group, respectively. In this sense, our approach is an instance of *geometric deep*

*learning* (Bronstein et al., 2021), and showcases how a combination of discrete and continuous symmetries can be exploited for addressing combinatorial optimization tasks.

The contributions of this work can be summarized as follows:

- We propose a novel deep learning model for lattice reduction outputting unimodular matrices via a *self-supervised objective.* The geometric architecture for the model incorporates *invariance and equivariance to continuous and discrete groups respectively.*

- We implement and empirically compare our model with the classical LLL algorithm for lattice reduction. We show that we can achieve comparable results in terms of complexity-performance. An intriguing insight from the result is that, hypothetically, without the knowledge of LLL, *one can employ self-supervised learning to discover a lattice reduction algorithm.*

- We extend the network to a convolutional model that *jointly reduces lattices arranged in a grid by leveraging the correlation between them.* In that sense, the overall cost of reduction can be amortized thanks to the correlation between lattices.

## 2 Related Work

**Deep Learning for Combinatorial Optimization.** Addressing combinatorial optimization problems via deep learning constitutes a growing line of research – see Bengio et al. (2021) for a comprehensive survey. In this context, models can be trained either by supervision on a dataset of optimal solutions (Vinyals et al., 2015; Prates et al., 2019; Lemos et al., 2019), or directly on the objective of the given combinatorial optimization problem (Toenshoff et al., 2019; Duan et al., 2022; Bello et al., 2016). Even though the latter raises differentiability challenges due to the discrete nature of the problems considered, it avoids data generation and allows the model to discover patterns autonomously. Our approach adheres to this paradigm since we train the model directly on the objective of lattice reduction. Moreover, combinatorial optimization problems often exhibit discrete symmetries – typically with respect to permutations. This has motivated the deployment of invariant and equivariant neural architectures, such as various incarnations of Graph Neural Networks (GNNs) – see Cappart et al. (2023) for an overview. On this note, the geometric nature of lattice reduction enables us to incorporate both continuous and discrete symmetries into our model.

**Applications of Lattice Reduction.** Finding closest or shortest vectors in a high-dimensional lattice is crucial in several domains. For example, a popular public-key cryptographic protocol is based on the hardness of the shortest vector problem (SVP) (Ajtai, 1996; Hoffstein et al., 1998; Regev, 2009). Lattice reduction also has some interesting applications in number theory, see Simon (2010). Moreover, several practical applications rely on the closest vector problem: probably, the most prominent one is the multiple-input multiple output (MIMO) detection in wireless communication, which is a crucial component of existing wireless technologies (5G, WiFi, etc.). An instance of integer least square, the problem is about finding an integer vector of transmitted symbols transmitted from multiple antenna. There is a vast literature around the topic - for example see (Yang & Hanzo, 2015; Wubben et al., 2004; Hassibi & Vikalo, 2005; Worrall et al., 2022; Taherzadeh et al., 2007) as examples. Another application is carrier phase estimation in the Global Positioning System (Hassibi & Boyd, 1998). Since lattice reduction significantly mitigates the computational cost of lattice-based problems, it has seen extensive deployment in all of these domains.

## 3 Background

### 3.1 Lattices

Intuitively, lattices are grids in arbitrary dimensions. More formally:

**Definition 3.1.** An $n$-dimensional *lattice* is a discrete subgroup of $(\mathbb{R}^n, +)$ of maximal rank.

If $\Lambda \subseteq \mathbb{R}^n$ is a lattice then there exists an isomorphism of groups $\Lambda \simeq \mathbb{Z}^n$. Such an isomorphism determines a *basis* of the lattice i.e., a set of linearly-independent vectors $b_1, \cdots, b_n \in \mathbb{R}^n$ such that $\Lambda = \mathbb{Z}b_1 \oplus \cdots \oplus \mathbb{Z}b_n$.

A basis is succinctly described by an invertible matrix $B \in \mathrm{GL}_n(\mathbb{R})$ whose columns are the basis vectors. Any two bases $B, B' \in \mathrm{GL}_n(\mathbb{R})$ are related via (right) multiplication by an integral invertible matrix i.e., $B' = BQ$ for some

$$Q \in \mathrm{GL}_n(\mathbb{Z}) = \{Q \in \mathbb{Z}^{n \times n} \mid \det(Q) = \pm 1\}. \tag{1}$$

Matrices belonging to $\mathrm{GL}_n(\mathbb{Z})$ are deemed *unimodular*.

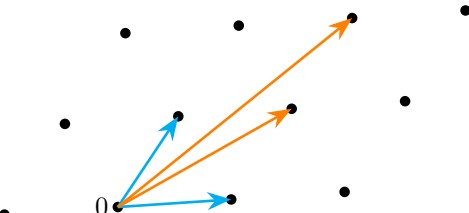

Figure 2: Two equivalent bases of a two-dimensional lattice.

Lattices carry two fundamental computational problems.

**Problem 3.2** (Shortest Vector Problem (SVP)). *Given a basis $B$ of a lattice $\Lambda$, find $\lambda \in \Lambda \setminus \{0\}$ minimizing the norm $\|\lambda\|$.*

**Problem 3.3** (Closest Vector Problem (CVP)). *Given a basis $B$ of a lattice $\Lambda$ and $x \in \mathbb{R}^n$, find $\lambda \in \Lambda$ minimizing the distance $\|\lambda - x\|$.*

The SVP reduces to the CVP, even in their respective approximate relaxations (Goldreich et al., 1999), and the SVP is known to be NP-hard under randomized reductions (Ajtai, 1998).

## 3.2 Lattice Reduction

Not all bases are created equal. The ideal bases are the *orthogonal* ones i.e., such that $B^T B$ is diagonal. However, not all lattices admit orthogonal bases, and the amount by which a basis is not orthogonal is measured by the following quantity.

**Definition 3.4.** The *orthogonality defect* of $B \in \mathrm{GL}_n(\mathbb{R})$ is

$$\delta(B) = \frac{\prod_i \|b_i\|}{|\det(B)|}. \tag{2}$$

Indeed, $\delta(B) \geq 1$ by Hadamard's inequality and $\delta(B) = 1$ if, and only if, $B^T B$ is diagonal. The denominator of Equation 2 is the volume of the fundamental domain of $\Lambda$ (i.e., the parallelepiped having the basis vectors as edges), and therefore it is independent of the basis.

The orthogonality defect is closely related to the SVP and the CVP. Indeed, the more orthogonal the given basis is, the easier these lattice problems become. In order to illustrate this, note that if $B$ is orthogonal then the shortest vector belongs to the columns of $B$, while finding the closest vector to a given point reduces to rounding the coordinates of the latter. For a general $B$, it can be proven via the Minkowski's theorem (Nguyen, 2009) that the shortest vector lies within bounds given by orthogonality defect $\delta(B)$, implying that the SVP simplifies for highly-orthogonal lattices. Therefore, given a lattice, it is convenient to find its most orthogonal basis – an operation referred to as *reduction*. This translates into the following computational problem.

**Problem 3.5** (Lattice Reduction). *Given $B \in \mathrm{GL}_n(\mathbb{R})$, find $Q \in \mathrm{GL}_n(\mathbb{Z})$ minimizing $\delta(BQ)$.*

However, the above problem is again NP-hard and therefore unfeasible to solve directly. The celebrated *Lenstra–Lenstra–Lovász (LLL) algorithm* (Lenstra et al., 1982) is an approximate algorithm for lattice reduction that finds a basis $B' = QB$ of an arbitrary lattice with orthogonality defect bounded by $\delta(B') \leq 2^{n(n-1)/4}$. The algorithm requires $\mathcal{O}(n^4 \log \beta)$ elementary arithmetic operations, where $\beta = \max_i \|b_i\|$, and therefore has polynomial complexity (Galbraith, 2012). We discuss the details of the LLL algorithm in

the Appendix (Section A). Other approximate lattice reduction algorithms similar to LLL are the Korkine–Zolotarev algorithm (Korkine & Zolotareff, 1877), which is slower and has worse orthogonality defect bound, and the Seysen algorithm (Seysen, 1993), which reduces the dual lattice simultaneously. Similarly to LLL, all of these algorithms are based on specific heuristics, and therefore solve the lattice reduction problem approximately.

## 4 Method

Our goal is to deploy (geometric) deep learning in order to approximate lattice reduction. To this end, we design a model of the form

$$\varphi : \mathrm{GL}_n(\mathbb{R}) \to \mathrm{GL}_n(\mathbb{Z}), \tag{3}$$

where the input represents a basis $B$ of a lattice, while the output represents the base-change unimodular matrix $Q$. The objective of the model on a datapoint $B$ is minimizing the (logarithmic) orthogonality defect of the reduced basis $B' = B\varphi(B)$ i.e., the loss is:

$$\mathcal{L}(B, \theta) = \log \delta(B') = \tag{4}$$

$$= \sum_i \log \|b'_i\| - \log |\det(B')| \geq 0, \tag{5}$$

where $\theta$ represents the parameters of $\varphi$. As mentioned before, the determinant in the above expression is independent of the basis ($\det(B') = \det(B)$) and therefore does not need to be optimized. In the following, we discuss two challenges related to the design of the model: outputting unimodular matrices and incorporating the symmetries of lattice reduction. Figure 3 provides an overview of the method.

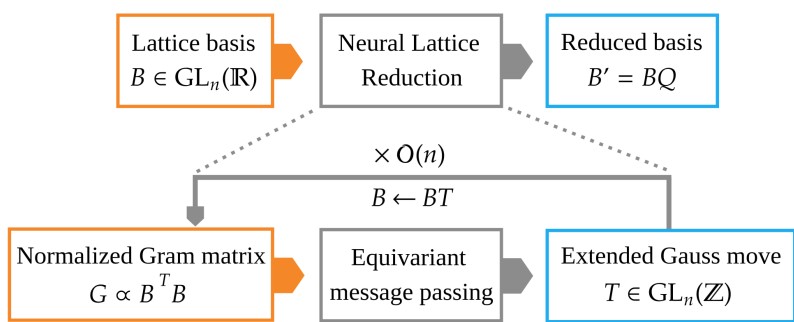

Figure 3: Overview of our method.

### 4.1 Unimodular Outputs

A first challenge is designing $\varphi$ in order to output unimodular matrices. This is subtle since the desired matrix has to be simultaneously discrete and invertible. To this end, we begin by considering the *special* subgroup $\mathrm{SL}_n(\mathbb{Z}) = \{Q \in \mathbb{Z}^{n \times n} \mid \det(Q) = 1\} \subseteq \mathrm{GL}_n(\mathbb{Z})$. Since the orthogonality defect is invariant to multiplying a basis vector by $-1$ (which flips the sign of the determinant of the basis matrix), without loss of generality we will focus on outputting an element of $\mathrm{SL}_n(\mathbb{Z})$. In the following, we will call *Gauss move* a $Q \in \mathrm{SL}_n(\mathbb{Z})$ with 1 on the diagonal ($Q_{i,i} = 1$ for all $i$) and at most one non-zero element $Q_{i,j}$ away from the diagonal ($i \neq j$). The following is a deep algebraic property of Gauss moves.

**Theorem 4.1** (Carter & Keller (1983)). *For $n \geq 3$, $\mathrm{SL}_n(\mathbb{Z})$ is boundedly generated by Gauss moves. Specifically, every matrix in $\mathrm{SL}_n(\mathbb{Z})$ is a product of at most $\frac{1}{2}(3n^2 - n) + 51$ Gauss moves.*

The above bound for the number of Gauss moves is not strict, and the optimal one is not known. Based on Theorem 4.1, we propose to design $\varphi$ to output a Gauss move and apply the model recursively to produce a sequence of moves. That is, given a priorly chosen $k \in \mathbb{N}$, the model $\psi$ produces a unimodular matrix $Q$ as:

$$Q = T_1 \cdots T_k, \qquad T_i = \varphi(BT_1 \cdots T_{i-1}), \tag{6}$$

where $T_i$ is a Gauss move.

Since the output space of a neural network is Euclidean, it is non-trivial to produce a Gauss move. To this end, we propose the following procedure. First, $\varphi$ outputs a matrix $M \in \mathbb{R}^{n \times n}$. The absolute value of its $n(n-1)$ entries away from the diagonal are interpreted as logits for a probability distribution over indices $(i, j)$ with $i \neq j$. An index $(i, j)$ is then sampled via the Gumbel-Softmax trick (Jang et al., 2016), which is necessary to differentiate through sampling. The corresponding entry $m_{i,j}$ of $M$ is used to build a Gauss move with all the non-diagonal entries apart from $m_{i,j}$ vanishing. Finally, $m_{i,j}$ is discretized in order to lie in $\mathbb{Z}$ in a differentiable manner. To this end, we adopt the procedure of *stochastic rounding*, which is popular in the network quantization literature (Gupta et al., 2015; Louizos et al., 2018). It consists of rounding $m_{i,j}$ to one of the two closest integers by sampling from a Bernoulli distribution with probability equal to (one minus) the rounding error. The sampling is performed via the Gumbel-Softmax trick (technically, Gumbel-Sigmoid), making it differentiable. This discretization procedure can be seen as an unbiased but stochastic version of the straight-through estimator (Yin et al., 2019).

The above procedure produces one Gauss move at a time. This is however inefficient, potentially requiring long sequences. According to the (loose) bound from Theorem 4.1, $k$ in Equation 6 should be chosen in the order of $\mathcal{O}(n^2)$. In order to improve this, we propose to retain an entire row and column of $M$ instead of a single entry. Specifically, after the index $(i, j)$ has been sampled, a matrix is produced with 1 on the diagonal, the $i$-th row and the $j$-th column equal to the ones of $M$ (except for the diagonal), and 0 elsewhere, i.e.:

$$
\begin{pmatrix}
1 & 0 & \cdots & & m_{1,j} & \cdots & 0 \\
0 & 1 & & & & & \\
\vdots & & \ddots & & \vdots & & \vdots \\
m_{i,1} & \cdots & 1 & \cdots & m_{i,j} & \cdots & m_{i,n} \\
\vdots & & \ddots & & \vdots & & \vdots \\
0 & & & \cdots & m_{n,j} & \cdots & 1
\end{pmatrix}
\tag{7}
$$

We refer to the above as an '*extended*' Gauss move. Extended Gauss moves enable $\varphi$ to output a unimodular matrix with $2n-3$ non-trivial entries simultaneously. This lowers the number of moves required to $\mathcal{O}(n)$, as shown below.

**Proposition 4.2.** *For $n \geq 3$, every matrix in $\mathrm{SL}_n(\mathbb{Z})$ is a product of at most $4n+51$ extended Gauss moves.*

See Appendix (Section B) for a proof. Again, the above bound is not only loose (especially for the additive constant), but might be mitigated on average for random lattices, as happens in practical scenarios. We provide an empirical investigation on the matter in Figure 5. In this case, the index $(i, j)$ is sampled via Gumbel-Softmax using the logits[1] matrix $R = |M|\mathbf{1} + \mathbf{1}|M| - |M|$, where $\mathbf{1}$ is the $n \times n$ matrix containing all 1s and $|M|$ is the entry-wise absolute value of the matrix $M$ (assuming the diagonal of $M$ to be zeroed-out beforehand).

Lastly, we remark that differentiating the recursive expression from Equation 6 w.r.t. (the parameters of) $\varphi$ can be computationally challenging, especially from a memory perspective. In order to circumvent this, we propose to detach the gradients of the network's output $T_i$ at each step $i$, and to optimize a cumulated loss $\sum_{i=1}^{k} \mathcal{L}(BT_1 \cdots T_i, \theta)$. As a consequence, the model learns to output an optimal Gauss move given the current input lattice, but the whole lattice reduction process is not optimized end-to-end.

## 4.2 Invariance and Equivariance

In this section, we discuss how to design $\varphi$ in order to satisfy invariance and equivariance properties according to the symmetries of the lattice reduction problem. We start by discussing the symmetry properties of reduced

---

[1]We have found important to compute the logits matrix $R$ from the matrix $M$ to achieve stable convergence.

bases. To this end, suppose that $B' = BQ$ is a basis minimizing the orthogonality defect (Equation 2) for some $B \in \mathrm{GL}_n(\mathbb{R})$. The following symmetries hold:

- If $\widetilde{B} = BH$ for some $H \in \mathrm{GL}_n(\mathbb{Z})$, then $B' = \widetilde{B}\widetilde{Q}$ minimizes the orthogonality defect for $\widetilde{B}$, where $\widetilde{Q} = H^{-1}Q$.

- If $\widetilde{B} = UB$ for some $U \in \mathrm{O}_n(\mathbb{R})$, then $\widetilde{B}Q$ minimizes the orthogonality defect for $\widetilde{B}$.

- If $\widetilde{B} = \alpha B$ for some $\alpha \neq 0 \in \mathbb{R}$, then $\widetilde{B}Q$ minimizes the orthogonality defect for $\widetilde{B}$.

The first above property comes from the fact that $B$ and $BH$ are bases of the same lattices and therefore have the same reduced basis. Instead, the second property is grounded in the intuition that $B$ and $UB$ are bases of lattices related by an isometry in the ambient space $\mathbb{R}^n$ and therefore can be reduced via the same base-change matrix $Q$. Similarly, scaling a lattice doesn't change its shape and the angle between its basis elements; invariance to scaling is evident in the definition of the orthogonality defect.

Based on the properties above, we aim to design a model $\varphi$ satisfying:

- *Right unimodular equivariance*: $\varphi(BH) = H^{-1}\varphi(B)$ for $H \in \mathrm{GL}_n(\mathbb{Z})$.

- *Left orthogonal invariance*: $\varphi(UB) = \varphi(B)$ for $U \in \mathrm{O}_n(\mathbb{R})$.

- *Scale invariance*: $\varphi(\alpha B) = \varphi(B)$ for $\alpha \neq 0 \in \mathbb{R}$.

We first focus on left orthogonal invariance. This is straightforward to achieve: instead of inputting the basis matrix $B$, we input the *Gram matrix* $G = B^T B$. The latter exhibits the desired invariance since, intuitively, it encodes intrinsic metric features of the lattice, which are preserved by isometries. In what follows, we will abuse the notation for $\varphi$ and write both $\varphi(B)$ and $\varphi(G)$, depending on the input considered.

Similarly, we address scale invariance by suitably normalizing the Gram matrix. A natural choice is normalizing by (the $n$-th root of) the *determinant* of the basis, which is an invariant of the lattice shared by any of its bases. Unfortunately, this quantity is often particularly small for lattices with a large defect and, therefore, it can produce extremely large input features, potentially posing challenges in training deep learning architectures. Instead, we opt for normalizing the Gram matrix by its *trace*.

In order to address right unimodular equivariance, note first that $G$ transforms as $BH \mapsto H^T GH$. Therefore, the model needs to satisfy $\varphi(H^T GH) = H^{-1}\varphi(G)$ for $H \in \mathrm{GL}_n(\mathbb{Z})$. However, the latter equivariance property turns our to be too challenging to be achieved. Not only it involves different transformations between input and output, but the unimodular group $\mathrm{GL}_n(\mathbb{Z})$ is algebraically subtle and understood only partially, due to its arithmetic nature. In order to circumvent this, we instead focus on a relevant finite subgroup of $\mathrm{GL}_n(\mathbb{Z})$: the *hyperoctahedral group* $\mathrm{H}_n$. The latter consists of the $2^n n!$ signed permutation matrices and can be thought as the group of isometries of both the hypercube and the cross-polytope. The hyperoctahedral group can also be interpreted as an arithmetic version of the orthogonal group, since $\mathrm{H}_n = \mathrm{GL}_n(\mathbb{Z}) \cap \mathrm{O}_n(\mathbb{R})$. Then, signed permutation matrices are intuitively not far from general unimodular ones: due to singular value decomposition, elements in $\mathrm{GL}_n(\mathbb{R})$ are obtained from the ones in $\mathrm{O}_n(\mathbb{R})$ by rescaling basis vectors in a given base. Therefore, equivariance to hyperoctahedral/orthogonal group is not far from equivariance to the full general group; the missing symmetries correspond to unisotropic rescalings. Even further, the (isotropic) *scale invariance* (third bullet point above) we impose in our model partially amends to this.

Considering the hyperoctahedral subgroup simplifies the challenge of designing an equivariant $\varphi$ for a number of reasons. First, $\mathrm{H}_n \subseteq \mathrm{O}_n(\mathbb{R})$ and therefore $H^{-1} = H^T$ for $H \in \mathrm{H}_n$. Moreover, the orthogonality defect is invariant to the hyperoctahedral group i.e., $\delta(BH) = \delta(B)$. Therefore, for the purpose of lattice reduction we can consider the equivariance property

$$\varphi(H^T GH) = H^T \varphi(G)H, \tag{8}$$

which is a natural tensorial transformation law and is compatible with the procedures described in Section 4.1 in the following sense. First, it accommodates the recursion from Equation 6 i.e., $Q$ inherits the equivariance property from $\varphi$. Second, the (stochastic) procedure to obtain extended Gauss moves is already equivariant (on average) in this sense i.e., if $M' = H^T M H$ then the corresponding Gauss move transforms accordingly.

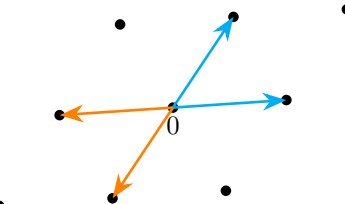

Figure 4: The hyperoctahedral group permutes and flips the sign of the basis vector.

In order to design a model that is equivariant as in Equation 8 for signed permutation matrices $H \in \mathrm{H}_n$, we opt for the Graph Neural Network (GNN) introduced by Maron et al. (2019). These networks can approximate arbitrary message-passing GNNs (Maron et al., 2018), which in turn are universal function approximators for functions over nodes of graphs (D'Inverno et al., 2024). In short, the computation performed by such a GNN is the following. The entries of the Gram matrix $G$ are interpreted as scalar features, which are then propagated by the GNN through a number of layers. Concretely, starting with the Gram matrix $G^1 = G \in \mathbb{R}^{n \times n \times 1}$, the $l$-th layer takes in input a tensor $G^l \in \mathbb{R}^{n \times n \times d_l}$ and produces the tensor $G^{l+1} = \left(G_i^{l+1}\right)_{i=1}^{i=d_{l+1}} \in \mathbb{R}^{n \times n \times d_{l+1}}$ given by:

$$G_i^{l+1} = \psi_1^l \left(G^l\right)_i \psi_2^l \left(G^l\right)_i + \psi_3^l \left(G^l\right)_i + \psi_4^l \left(G^l\right)_i^T, \tag{9}$$

where $\psi_1^l, \cdots, \psi_4^l \colon \mathbb{R}^{d_l} \to \mathbb{R}^{d_{l+1}}$ are Multi-Layer Perceptrons (MLPs) that are applied to $G^l$ entry-wise in the $n \times n$ indices. The MLPs are required to be sign-equivariant, which can be achieved by implementing sign-equivariant activation functions e.g., the soft-threshold $\sigma(x) = \mathrm{ReLU}(|x| - |b|) \, \mathrm{sign}(x)$, where $b$ is the bias of the MLP. After $L$ iterations, the GNN outputs $G^L \in \mathbb{R}^{n \times n \times 1}$, which is interpreted as the matrix $M$ from Section 4.1.

## 4.3 Computational Complexity

The computational cost of a forward pass in our model with respect to the dimension $n$ can be estimated as follows. First, computing the Gram matrix $G = B^T B$ has complexity $\mathcal{O}(n^3)$. Next, the message-passing procedure of the GNN layer (Equation 9) can be implemented in $\mathcal{O}(n^3)$ as well. Lastly, the procedure from Section 4.1 to produce extended Gauss moves is easily seen to have complexity $\mathcal{O}(n^2)$. Therefore, the overall complexity is $\mathcal{O}(n^3 k)$, where $k$ is the number of extended Gauss moves deployed (see Equation 6). The latter has to scale as $k = \mathcal{O}(n)$ according to the bound from Proposition 4.2. However, the bound is a (loose) worst-case estimate, and can therefore be mitigated – even asymptotically – on expectation for specific distributions of lattices, as happens in practice. Hence, the forward pass of our model is at least as efficient as the LLL algorithm asymptotically , which runs in $\mathcal{O}(n^4)$ (see Section 3.1). Still, the computational cost of deep networks includes other constants related to e.g. the number of channels, which can have important impact on the runtime for low values of $n$. Using large models is understood to be very effective for optimization, and there exist many techniques (such as pruning and knowledge distillation (Hinton, 2015; Gou et al., 2021)) to subsequently reduce the computational cost with minimal loss in performance. Specifically, it is possible to distil a model trained with a large architecture and with a large value of $k$ into another one with a small architecture, possibly factorizing the unimodular matrix $Q$ in fewer than $k$ matrices. The latter avoids running the model $k$ times at test time to produce the reduced lattice, amending for the computational cost inherent in training with a large $k$. In this work, we mostly focus on the overall performance of our models to prove their potential to tackle the lattice reduction problem and leave improving efficiency for future works.

### 4.4 Joint Lattices Reduction

So far, we have focused above on the neural lattice reduction for a single lattice. In practice, there are cases where the goal is to reduce a large number of correlated lattices. Although this can be done in parallel, the question is whether we can leverage the correlation between lattices to improve on performance-complexity trade-off.

**Application to MIMO detection.** A notable example of this scenario is multiple antenna wireless technology, called multiple-input multiple-output (MIMO). In wireless communication, time and frequency are two physical *resources* which are used to multiplex information. In this technology, the transmission occurs over multiple time-frequency blocks. In each block, a lattice point, say $\mathbf{x} \in \mathbb{C}^n$, constrained within a box in the underlying space, is sent using multiple antenna technology. The impact of electromagnetic propagation is captured via a linear transformation, say the channel matrix $B \in \mathrm{GL}_n(\mathbb{C})$. The signal detection, called MIMO detection problem, boils down to solving the integer least-square problem $\mathbf{y} = B\mathbf{x} + N$, where $N$ represents white noise and $B$ is assumed to be known. Among other methods for solving this problem, we can apply lattice reduction to the channel matrix $B$ (Wubben et al., 2004; Worrall et al., 2022). This step needs to be done for each time-frequency block and incurs a substantial computational complexity in the processing chain. For example, in 5G, the fundamental resource block consists of 168 blocks (14 time and 12 frequency blocks). For a transmission over 20 MHz bandwidth, in certain operation regimes, we need 106 of these blocks, which means 17808 parallel lattice reduction (Chen et al., 2021). However, the channel matrices share a strong correlation stemming from their adjacency in the time-frequency grid. This shared correlation over the grid can be leveraged to amortize some of the operations of our neural lattice reduction approach resulting in joint reduction of correlated lattices at a reduced cost. We review the basics of wireless communication more in Appendix D.

**Convolutional Neural Lattice Reduction** In order to leverage the spatial correlation between neighboring lattices, we adapt the previous architecture by 1) endowing its activations with a *local field of view* to catch the spatial correlation between the lattices and by 2) adopting an *hourglass design* to amortize the reduction cost over the whole grid.

More precisely, each linear layer in the MLP used to build the equivariant message passing in Eq. 9 is replaced with a convolution layer with the same input and output channels but with a $3 \times 3$ kernel size. Inspired by the U-Net architecture (Ronneberger et al., 2015), to reduce the computational cost, our model consists of a first encoder which heavily down-samples the input grid via strided-convolution and following decoder which up-samples the features to the original resolution using transposed convolution and nearest neighbor interpolation; we also adopt linear skip connections based on $1 \times 1$ convolution between feature maps at the same resolution to avoid losing information about the input lattices. Finally, a different extended Gauss move is predicted for each lattice in the grid by a last small equivariant MLP.

While this architecture is more expensive than the single-lattice architecture proposed in the previous section, its cost is shared among all lattices within the input grid and, therefore, the cost per lattice is potentially reduced.

## 5 Experiments

### 5.1 Lattice Reduction Under Different Distributions

In this section, we study the performance of our method when trained on lattices generated from different distributions. In particular, we consider 4 different datasets:

- `Uniform`. The entries of $B \in \mathbb{R}^{n \times n}$ are i.i.d. and sampled from the uniform distribution $b_{ij} \sim U(0,1)$

- `Exponential`. We sample a matrix $B \in \mathrm{GL}_n(\mathbb{R})$ by first sampling a matrix $B' \sim$ `Uniform` and, then, applying the matrix exponential $B = \exp(0.5 \cdot B')$.

| $n$ | Distribution | Initial $\log \delta(B)$ | LLL $\log \delta(B'_{\text{LLL}})$ | MLP $\log \delta(B'_{\text{MLP}})$ | Our $\log \delta(B'_{\text{our}})$ | Gap (%) $\frac{\log \delta(B'_{\text{our}}) - \log \delta(B'_{\text{LLL}})}{\log \delta(B)} \cdot 100\%$ |
|---|---|---|---|---|---|---|
| 4 | `Uniform` | 3.46 $_{(1.33)}$ | 0.16 $_{(0.08)}$ | 0.23 $_{(0.27)}$ | 0.18 $_{(0.17)}$ | 0.38 $_{(6.10)}$ |
|   | `Exponential` | 0.98 $_{(0.30)}$ | 0.30 $_{(0.10)}$ | 0.31 $_{(0.13)}$ | 0.28 $_{(0.09)}$ | $-2.91$ $_{(15.5)}$ |
| 6 | `Uniform` | 5.74 $_{(1.41)}$ | 0.48 $_{(0.15)}$ | 0.79 $_{(0.61)}$ | 0.52 $_{(0.22)}$ | 0.73 $_{(5.0)}$ |
|   | `Convex` $d{=}3$ | 7.17 $_{(1.37)}$ | 0.48 $_{(0.14)}$ | 0.85 $_{(0.59)}$ | 0.51 $_{(0.20)}$ | 0.36 $_{(3.9)}$ |
|   | `Ajtai` $q{=}8$ | 6.56 $_{(4.39)}$ | 0.48 $_{(0.15)}$ | 3.51 $_{(2.16)}$ | 0.55 $_{(0.32)}$ | 1.3 $_{(8.6)}$ |
| 8 | `Uniform` | 8.02 $_{(1.42)}$ | 1.00 $_{(0.23)}$ | 2.78 $_{(1.44)}$ | 1.16 $_{(0.47)}$ | 2.05 $_{7.04}$ |
|   | `Exponential` | 6.29 $_{(0.93)}$ | 1.05 $_{(0.20)}$ | 1.41 $_{(0.45)}$ | 1.04 $_{(0.21)}$ | $-0.27$ $_{(4.78)}$ |
|   | `Convex` $d{=}3$ | 9.78 $_{(1.34)}$ | 1.01 $_{(0.27)}$ | 2.17 $_{(0.99)}$ | 1.18 $_{(0.48)}$ | 1.72 $_{(5.72)}$ |
|   | `Convex` $d{=}8$ | 13.0 $_{(1.36)}$ | 0.80 $_{(0.22)}$ | 3.33 $_{(1.58)}$ | 0.95 $_{(0.49)}$ | 1.97 $_{(4.13)}$ |
|   | `Convex` $d{=}16$ | 14.8 $_{(1.37)}$ | 0.82 $_{(0.22)}$ | 2.47 $_{(1.16)}$ | 1.00 $_{(0.52)}$ | 1.29 $_{(3.81)}$ |
|   | `Ajtai` $q{=}8$ | 9.83 $_{(6.35)}$ | 1.00 $_{(0.23)}$ | 9.83 $_{(6.35)}$ | 1.32 $_{(0.75)}$ | 4.2 $_{(11.7)}$ |

Table 1: Final log orthogonality defect achieved by LLL, an MLP baseline and our method on different lattice distributions. The table reports the average performance (and standard deviation) on 4000 lattices used for validation.

- `Convex`. We first sample $d \ll n^2$ lattice bases $\{B'_i \sim \texttt{Uniform}\}_i^d$. Then, a lattice basis is sampled as a random convex linear combination of these $d$ bases as $B = \sum_i^d \frac{w_i}{\sum_j w_j} B'_i$, with $w_i \sim U(0,1)$.

- `Ajtai`. Inspired by Ajtai (1996), we sample a lattice basis $B'$ with i.i.d. entries sampled uniformly in $\{\frac{0}{q}, \frac{1}{q}, \cdots, \frac{q-1}{q}\}$ and then use the corresponding dual lattice, with basis $B = B'^{-T}$.

A larger dimension $n$ generally indicates harder datasets. The matrix exponential can increase the orthogonality defect by introducing more correlation between different entries; however, the 0.5 factor used in the `Exponential` distribution attenuates this effect, generally leading to more orthogonal lattices than `Uniform`. The `Convex` distributions aim to simulate a form of '*manifold hypothesis*', where the data lives in a much lower dimension $d \ll n^2$ than the ambient feature space. With this distribution, we want to verify if our method can be fine-tuned on particular data distributions - where deep learning solutions typically shine - to perform better than a general-purpose algorithm designed to work well in the average scenario. Finally, the dual lattices in the `Ajtai` distribution should not be harder to reduce than the original lattices; because the matrix $B'$ has a distribution similar to `Uniform`, these two datasets are expected to have a similar complexity.

In these experiments, we generate the training data procedurally by sampling random matrices from the distributions above. For evaluation, we sample once 4000 lattices from each distribution and use them to evaluate both our method and LLL. Additionally, to facilitate optimization, we find useful to increase the number of steps $k$ gradually up to $k = 2n$ during the training iterations.

Table 1 compares the performance of our method, trained on each distribution independently, with that of LLL (using the standard parameter $\delta = 0.75$) and of an MLP baseline[2] . First, note that the performance of LLL is approximately constant over the `Uniform`, `Convex` and `Ajtai` distributions; these results are in line with the observations in the previous paragraph about the similar complexity of these datasets. Overall, we observe that our method approaches the performance of the classical algorithm in most settings, as indicated by the low relative performance *gap* in Table 1. However, with respect to LLL, our method seems to improve in the `Exponential` setting but struggles more to find good solutions for the `Ajtai` lattices.

Indeed, the `Exponential` distribution turns out to be slightly harder for LLL, despite the initial lower defect; we note that the initial defect is not necessarily a good indicator of the complexity of the reduction, and we hypothesize that the matrix exponential still introduces certain additional correlations. Regarding the

---

[2]The MLP model adopts an architecture similar to our method, i.e. it samples $k$ extended Gauss moves with the same recursive strategy, but it replaces the equivariant layers with unconstrained linear layers and treats the $n^2$ entries of the Gram matrix as a generic input feature vector.

`Ajtai` lattices, while they are not theoretically harder to reduce, the inverse matrix $B'^{-T}$ often contains a wider range of values (occasionally up to $10^5$) – especially since $B'$ takes values in $[0, 1]$ – resulting in potential challenges for a neural network, which justifies the poorer performance found. In the experiment with the `Convex` distribution, we observe that the gap between our method and LLL slightly decreases for lower values of $d$. In addition, the consistent performance gap between the MLP baseline and our method across all datasets in Table 1 indicates the importance of adopting an equivariant design to efficiently learn solutions to this combinatorially hard problem.

Finally, Figure 5 shows how the number of steps $k$ affects the performance of our models on the `Uniform` datasets. In particular, we observe an asymptotic convergence towards the LLL performance as the number of steps increases, suggesting the performance gap could be closed provided enough steps are employed. Additionally, while using a larger number steps keeps improving performance monotonically, we observe a quick saturation after approximately $k = n$ steps. This observation is consistent with Proposition 4.2, which establish an upper-bound of $O(n)$ steps to generate any unimodular matrix. Hence, we consider it plausible that $O(n)$ steps are generally sufficient to achieve a sufficiently low performance gap with respect to LLL.

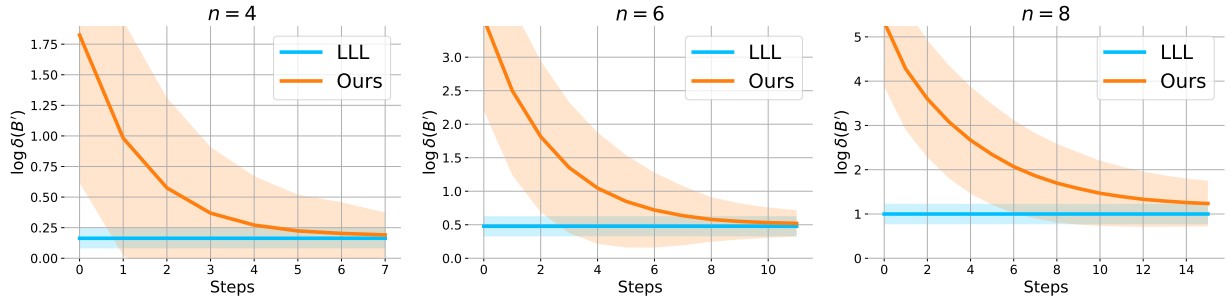

Figure 5: Performance of our method at different steps vs LLL on $n$ dimensional lattices from the `Uniform` distribution. The trends indicate that the performance of our models starts to saturate after about $n$ steps; this is in line with the theoretical upper-bound of $O(n)$ steps in Proposition 4.2.

## 5.2 Joint Reduction of Correlated Lattices

In this section, we present results for joint reduction of correlated lattices arranged in a grid, which has applications for MIMO detection as explained in Section 4.4.

For data generation in our experiments, we used a 5G-compliant physical downlink shared channel (PDSCH) simulator implemented with MATLAB's 5G Toolbox (Inc., 2022b;a). We used the standard wireless channel models to simulate the communication pipeline. For the sake of simplicity, we assume perfect channel estimation at the receiver. To get the perfect channel estimate, we used MATLAB's `nrPerfectChannelEstimate` function. We simulated tapped delay line (TDL) channel model with TDLA30 delay profile and a Doppler shift of 100 Hz. This model represents a statistical model for the behavior that can be attributed to an indoor propagation environment. We simulated $2 \times 2$ and $4 \times 4$ MIMO system over a time-frequency grid of size $14 \times 48$ (i.e. 4 resource blocks stacked along the frequency axis). Since the channel matrices are complex valued, we are dealing with *real-valued* lattice basis of dimension $4 \times 4$ and $8 \times 8$ for each time-frequency block in the $14 \times 48$ grid. For more details see Appendix D.3.

Additionally, the MIMO detection problem is more challenging when the channel matrix $B$ does not yield an orthogonal lattice. This happens because of the arrangements of transmit and receive antenna panels and characteristics of the propagation environment. In statistical models used for wireless communication, this effect is modelled by creating statistical correlations between rows and columns of the matrix $B$ - see 3rd Generation Partnership Project (3GPP); Heath & Lozano (2019). Hence, we consider datasets generated under three correlations regimes (`Low`, `Medium` and `High`) using standard channel models for our evaluations.

In Table 2, we compare the convolutional model described in Section 4.4 with an LLL baseline. The baseline uses LLL to reduce each of the $14 \times 48$ lattices within a grid independently. Conversely, the encoder of our

| $n$ | Correlation | Initial | LLL | Our |
|---|---|---|---|---|
| 4 | Low | 0.99 (1.00) | 0.185 (0.170) | 0.133 (0.128) |
| | Medium | 2.38 (1.46) | 0.117 (0.100) | 0.123 (0.126) |
| | High | 3.96 (1.89) | 0.095 (0.105) | 0.131 (0.163) |
| 8 | Low | 2.99 (1.37) | 1.05 (0.31) | 1.01 (0.43) |
| | Medium | 20.6 (3.34) | 0.91 (0.34) | 4.14 (1.66) |
| | High | 23.4 (4.11) | 0.56 (0.37) | 6.36 (2.48) |

Table 2: Performance of our convolutional model and LLL on the wireless channels lattices at different correlations levels and dimensions. The table reports the average performance (and standard deviation) on 10000 grids (each containing $12 \times 48$ correlated lattices) used for validation.

network down-samples the input grid to a $3 \times 12$ and, then, to a $1 \times 3$ grid; finally, the decoder up-samples the intermediate features to the original resolution. During the training iterations, we gradually increase the number of steps $k$ up to $n + 1$. For each experimental setting, we use a fixed training set of 20000 grids and evaluate the methods on a fixed set of 10000 grids.

Overall, our method outperforms the LLL baseline on the `Low` correlation datasets and achieves comparable performance on the other $n = 4$-dimensional datasets. Unfortunately, in the higher correlation $n = 8$-dimensional datasets, our model fails to match the LLL performance.

## 6  Conclusions, Limitations and Future Work

In this work, we have addressed lattice reduction via deep learning methods. To this end, we have designed a deep neural model outputting factorized unimodular matrices and trained it in a self-supervised manner. We have incorporated the symmetries of lattice reduction into the model by making it invariant with respect to the orthogonal group and scaling and equivariant with respect to the hyperoctahedral group. We also proposed an architecture to jointly reduce correlated lattices and applied it on wireless channel data.

While our neural method does not consistently outperform the popular LLL algorithm, this is the first work that demonstrates the *feasibility* and *effectiveness* of a deep learning approach to solve the NP-hard lattice reduction problem. The benefit of this approach is that it can be tuned on specific data domains and that it can perform *amortized reduction* of multiple correlated lattices. Endowing classical algorithms for combinatorial optimization problems with these desirable properties is a general and challenging program which finds applications in several areas. Hence, we hope that our work will inspire more research on applying neural networks to lattice problems.

Finally, our model is not equivariant (on the right) to the whole unimodular group but only to the hyperoctahedral subgroup. Therefore, we do not leverage upon the complete spectrum of symmetries of the lattice reduction problem. The challenge of designing a model equivariant to $\mathrm{GL}_n(\mathbb{Z})$ remains open, and represents another interesting line for future investigation.

## 7  Acknowledgements

We wish to thank Daniel Worral, Roberto Bondesan and Markus Peschl for their help and the insightful discussions.

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

# A The LLL Algorithm

In this section, we describe the Lenstra–Lenstra–Lovász (LLL) algorithm for lattice reduction (Lenstra et al., 1982) in details. Let $B \in \mathrm{GL}_n(\mathbb{R})$ be a basis of a lattice and denote by $B^*$ the orthogonal basis of $\mathbb{R}^n$ obtained via the Gram-Schmidt algorithm applied to $B$. For each $i, j$ consider moreover

$$\mu_{i,j} = \frac{b_i \cdot b_j^*}{\|b_j^*\|^2}. \tag{10}$$

**Definition A.1.** A basis $B \in \mathrm{GL}_n(\mathbb{R})$ is *Siegel-reduced* if for all $i > j$:

$$|\mu_{i,j}| \leq \tfrac{1}{2} \qquad\qquad \frac{\|b_i^*\|^2}{\|b_{i-1}^*\|^2} \geq \left(\tfrac{3}{4} - \mu_{i,i-1}^2\right)$$

$$\textit{Size Condition} \qquad\qquad \textit{Lovász Condition}$$

The following is a bound on the orthogonality defect for Siegel-reduced bases.

**Lemma A.2** (Lenstra et al. (1982)). *If $B$ is Siegel-reduced then:*

$$\delta(B) \leq 2^{\frac{n(n-1)}{4}}. \tag{11}$$

The LLL algorithm finds a Siegel-reduced basis of an integral lattice $\Lambda \subseteq \mathbb{Z}^n \subseteq \mathbb{R}^n$ starting from an arbitrary basis $B \in \mathrm{GL}_n(\mathbb{R}) \cap \mathbb{Z}^{n \times n}$ (see Algorithm 1). It can be seen that the algorithm requires $\mathcal{O}(n^4 \log \beta)$ elementary arithmetic operations, where $\beta = \max_i \|b_i\|$, and therefore has polynomial complexity (Corollary 17.5.4 in Galbraith (2012)).

---

**Algorithm 1** LLL Algorithm

---

**Require:** Basis of an (integral) lattice $B \in \mathrm{GL}_n(\mathbb{R}) \cap \mathbb{Z}^{n \times n}$
**Ensure:** Siegel-reduced basis $B$
  $B^* \leftarrow \text{Gram-Schmidt}(B)$
  $k \leftarrow 2$
  **while** $k \leq n$ **do**
    **for** $j = k - 1$ to $1$ **do**
      **if** $|\mu_{k,j}| > \tfrac{1}{2}$ **then**
        $b_k \leftarrow b_k - \lceil \mu_{k,j} \rceil b_j$
        $B^* \leftarrow \text{Gram-Schmidt}(B)$
      **end if**
    **end for**
    **if** $\frac{\|b_k^*\|^2}{\|b_{k-1}^*\|^2} \geq \left(\tfrac{3}{4} - \mu_{k,k-1}^2\right)$ **then**
      $k \leftarrow k + 1$
    **else**
      Swap $b_k$ and $b_{k-1}$
      $k \leftarrow \max\{k - 1, 2\}$
      $B^* \leftarrow \text{Gram-Schmidt}(B)$
    **end if**
  **end while**

---

# B    Theoretical Results

## B.1    Proof of Main Result

**Proposition B.1.** *For $n \geq 3$, every matrix in $\mathrm{SL}_n(\mathbb{Z})$ is a product of at most $4n+51$ extended Gauss moves.*

*Proof.* Pick $Q \in \mathrm{SL}_n(\mathbb{Z})$. Note first that the non-zero entries of each row and each column of $Q$ are coprime since they can be arranged in an integral linear combination equal to $\det(Q) = 1$. Next, consider the last row of $Q$, denoted by $u_1 = Q_{n,1}, \cdots, u_n = Q_{n,n}$. We would like to show that the non-zero entries among $u_1, \cdots, u_{n-1}$ can be made coprime by multiplying $Q$ on the right by at most one (non-extended) Gauss move. If $u_n = 0$, this is true already for $Q$ by the above observation. Suppose that $u_n \neq 0$. If there are less than two indices $1 \leq i < n$ such that $u_i \neq 0$, we can replace one vanishing $u_i$ with $u_n$ by multiplying $Q$ on the right by a (non-extended) Gauss move and obtain the desired result. Otherwise, assume without loss of generality that $u_1 \neq 0$ and consider some $t \in \mathbb{Z}$ such that:

- $t \equiv 1$ mod all the primes dividing all the non-zero $u_i$ for $i \in \{1, \cdots, n-1\}$,

- $t \equiv 0$ mod all the primes dividing all the non-zero $u_i$ for $i \in \{2, \cdots, n-1\}$ but not dividing $u_1$.

Then the non-zero integers among $u_1 + tu_n, u_2 \cdots, u_{n-1}$ are coprime, as desired.

We therefore assume that $u_1, \cdots, u_{n-1}$ are coprime. By the Bézout's identity, there exist integers $a_1, \cdots, a_{n-1}$ such that $\sum_i a_i u_i = 1 - u_n$. By multiplying $Q$ on the right by the extended Gauss move

$$
\begin{pmatrix}
1 & 0 & \cdots & a_1 \\
0 & 1 & & \vdots \\
\vdots & & \ddots & a_{n-1} \\
0 & \cdots & 0 & 1
\end{pmatrix}
\tag{12}
$$

we reduce to the case $u_n = 1$. By further multiplying on the right by the extended Gauss move

$$
\begin{pmatrix}
1 & 0 & \cdots & 0 \\
0 & 1 & & \vdots \\
\vdots & & \ddots & 0 \\
-u_1 & \cdots & -u_{n-1} & 1
\end{pmatrix}
\tag{13}
$$

we reduce to the case where the last row is $(0, \cdots, 0, 1)$. Similarly, by multiplying by another move on the left, the last column reduces to $(0, \cdots 0, 1)$, obtaining a matrix of the form:

$$
\begin{pmatrix}
 & & & 0 \\
 & A & & \vdots \\
 & & & 0 \\
0 & \cdots & 0 & 1
\end{pmatrix}
\tag{14}
$$

where $A \in \mathrm{SL}_{n-1}(\mathbb{Z})$.

Putting everything together, via induction with $4n - 12$ extended Gauss moves we arrive at a matrix of the form $A' \oplus I$, where $A' \in \mathrm{SL}_3(\mathbb{Z})$ and $I$ is the identity matrix of dimension $n - 3$. Since by Carter & Keller (1983) any matrix in $\mathrm{SL}_3(\mathbb{Z})$ can be written as a product of 63 (non-extended) Gauss moves, this concludes the proof.

$\square$

## B.2    Constructing Equivariant Maps

In this section, we explore classes of maps that obey an equivariance property closely related to our model. In particular, we prove an impossibility result for multi-linear maps. To this end, recall that the input of our

model is an (invertible) matrix, which can be seen as an element of the representation $\mathbb{R}^{n \times n}$ of $\mathrm{GL}_n(\mathbb{Z})$ on which the latter acts by right matrix multiplication. Therefore, it is convenient to describe the equivariant maps in this context.

**Proposition B.2.** *Consider a $\mathrm{GL}_n(\mathbb{Z})$-equivariant map $\varphi : \mathbb{R}^{n \times n} \to \mathbb{R}^{n \times n}$. Then there exists a map $f : \mathbb{R}^{n \times n} \to \mathbb{R}^n$ such that the $i$-th column of $\varphi(B)$ is given by*

$$f(b_i, \cdots, b_n, b_1, \cdots, b_{i-1}) \tag{15}$$

*where $b_*$ denotes the corresponding column of $B$. Moreover, $f$ satisfies the following two conditions for all $B = (b_1, \cdots, b_n)$:*

- $f(b_1, \cdots, b_n) = f(b_1, b_2, \cdots, b_i, b_i + b_{i+1}, b_{i+2}, \cdots)$ *for all $1 \leq i < n$,*

- $f(b_1, \cdots, b_n) + f(b_n, b_1, \cdots, b_{n-1}) = f(b_1 + b_n, b_2, \cdots, b_n)$.

*Any equivariant map $\varphi$ is obtained from an $f$ satisfying the above conditions.*

*Proof.* It is well-known that $\mathrm{GL}_n(\mathbb{Z})$ is generated as a group by the following two matrices (Newman, 1972):

$$S = \begin{pmatrix} 0 & 0 & 0 & & 0 & 1 \\ 1 & 0 & 0 & \cdots & 0 & 0 \\ 0 & 1 & 0 & & 0 & 0 \\ \vdots & & \ddots & & \vdots \\ 0 & 0 & 0 & \cdots & 1 & 0 \end{pmatrix} \qquad T = \begin{pmatrix} 1 & 1 & 0 & & 0 & 0 \\ 0 & 1 & 0 & \cdots & 0 & 0 \\ 0 & 0 & 1 & & 0 & 0 \\ \vdots & & \ddots & & \vdots \\ 0 & 0 & 0 & \cdots & 0 & 1 \end{pmatrix} \tag{16}$$

Given a matrix $B \in \mathbb{R}^{n \times n}$, $BS$ cycles the columns of $B$ from right to left, while $BT$ replaces the second column with itself summed with the first one. Now, pick a map $\varphi : \mathbb{R}^{n \times n} \to \mathbb{R}^{n \times n}$ and consider its column-wise components $\varphi_i$ i.e., $\varphi(B)_i = \varphi_i(B)$, where the subscript denotes the column of the matrix. Equivariance w.r.t. to $S$ is equivalent to $\varphi_i(B)$ coinciding with $\varphi_1$ computed on $B$ with its columns cycled by $i$ steps. This implies the existence of $f$ as in the statement. Equivariance w.r.t. $T$ is then equivalent to the two conditions in the statement. □

Therefore, the problem of classifying equivariant maps reduces to solving the functional equations for $f$ from Proposition B.2. Some solutions can be found by forcing $f$ to be linear and dependent only on the first column i.e., $f(B) = f(b_1)$ by abuse of notation, with $f$ linear. These are all the maps that are equivariant to the whole $\mathrm{GL}_n(\mathbb{R})$. The next step is looking for a *multi-linear* $f$ which is not dependent on the first column alone. In this case the two conditions translate in the following ones:

- $f(b_1, b_2, \cdots, b_i, b_i, b_{i+1}, \cdots, b_n) = 0$ for all $1 \leq i < n$,

- $f(b_n, b_1, \cdots, b_{n-1}) = f(b_n, b_2, \cdots, b_n)$.

The second condition above implies inductively that $f = 0$, meaning that *there are no non-trivial multi-linear equivariant maps.*

## C Implementation Details

We implement the model described by Equation 9, where $\psi_j^l$ are simple MLPs which include a single linear layer (without bias) followed by a soft-threshold activation $\sigma(x) = \text{ReLU}(|x| - |b|) \cdot \text{sign}(x)$.

Because the $n$ elements on the diagonal of the Gram matrix and of the intermediate features $G^l$ are sign-invariant, we also add to Equation 9 a simple message passing layer between them similar to Deep-Set: $[G^{l+1}]_{ii} + = \psi_5^l([G^l]_{ii}) + \frac{1}{n} \sum_j \psi_6^l([G^l]_{jj})$, where $[G^l]_{ii}$ is the $i$-th element in the diagonal of $G^l$ while $\psi_5^l$ and $\psi_6^l$ are simple MLPs, including a normal linear layer followed by a standard ReLU activation.

The single lattice architecture includes 4 message passing layers and a final 2-layers MLP processing the features of each of the $n^2$ entries to output the extended Gauss move. All intermediate features use 64 channels for each of the $n^2$ entries.

In the convolutional architectures, downsampling is performed in two message passing layers which employ convolution with stride 4; upsampling is also performed in two steps by interleaving two message passing layers using transposed convolution with stride 2 and two nearest neighbors interpolations. All intermediate features use at most 10 channels for each of the $n^2$ entries in each pixel in the grid.

To improve expressiveness, we augment the input features with a matrix containing the projection coefficients $p_{ij} = \frac{b_i^T b_j}{|b_j|_2^2}$ and a matrix containing their quantization $\lfloor p_{ij} \rceil$.

Moreover, in order to mitigate the instability due to discretization when sampling an extended Gauss move, we include a normalization layer (Ba et al., 2016) in the second-last activations. The number of extended Gauss moves is set $k = 2n$ in the single lattice experiments and $k = n + 1$ in the joint lattice reduction ones. Training is performed via the Adam optimizer (Kingma & Ba, 2014).

# D  Lattice Reduction in Wireless Communication

We review the basic building blocks in a conventional wireless communication system and discuss where lattice reduction fits into the problem.

## D.1  Wireless Processing Chains

The transmitter maps the binary data to the constellation points using a modulator block that exploits a certain modulation scheme, such as Quadrature Amplitude Modulation (QAM). The most frequently used QAM constellations consists of points arranged in a square grid with equal vertical and horizontal spacing, for e.g., 4-QAM, 16-QAM, and 64-QAM, see Fig. 6.

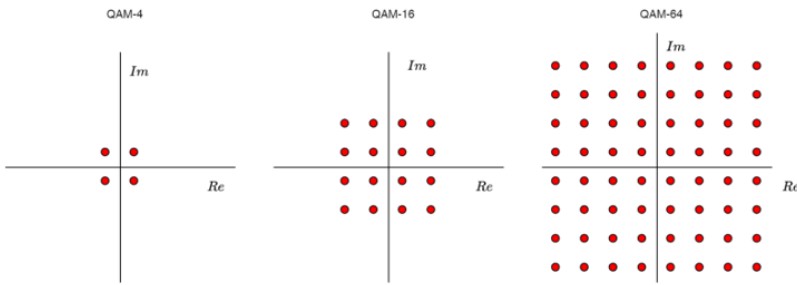

Figure 6: An example of QAM constellations

The QAM symbols at time slot $t$ are mapped to individual resources on the resource grid as in Fig. 7.

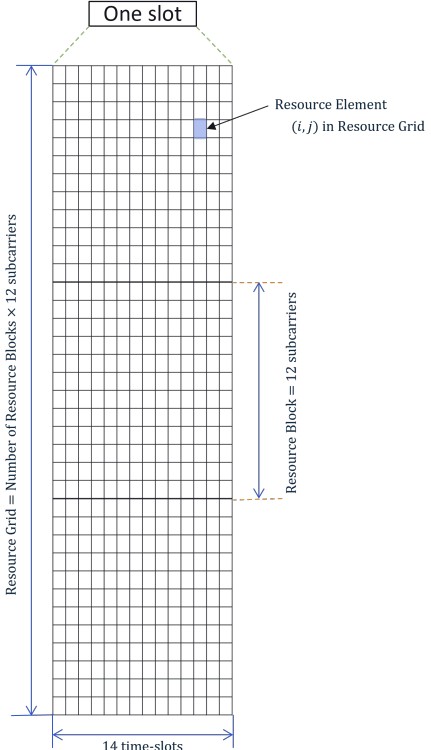

Figure 7: An example of time-frequency resource grids used in wireless technology which includes three resource blocks.

In the case of MIMO, symbols from QAM constellation $\Omega$ gets mapped to each of the individual resources on the grid, which are then transmitted over $n_t$ antennas on the transmit side. Fig. 8 depicts a $2 \times 2$ MIMO system, where $\mathbf{x} \in \Omega^2$ represents the transmit vector for an individual resource element, consisting of QAM constellation symbols pertaining to each of the transmit antennas.

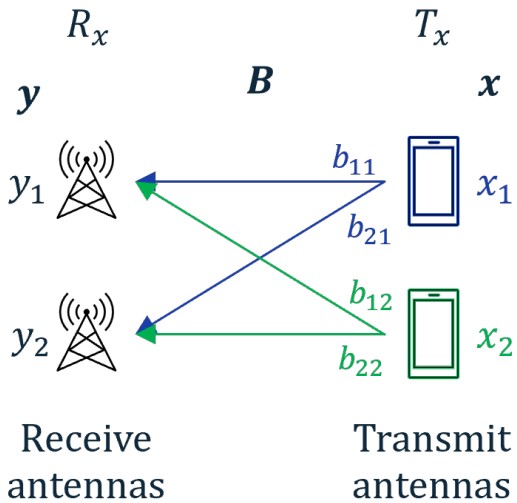

Figure 8: An example of MIMO communication system

## D.2 Lattice Reduction for MIMO Detection

Consider a MIMO system with $n_t$ transmit and $n_r$ receive antennas. For simplicity, we assume $n_t = n_r = n$. The transmit vector is the modulated bit-stream using a constellation $\Omega$ (see above). Therefore, the transmit vector is given by $\mathbf{x} \in \Omega^n$. The wireless processing chain maps $\mathbf{x}$ to a signal that is transmitted over $n$ antennas. On the receiver side, after some processing, the effect of the channel can represented by the channel matrix $B$[3], which relates the observed vector $\mathbf{y}$ to $\mathbf{x}$ as:

$$\mathbf{y} = B\mathbf{x} + N,$$

where $N$ is the additive noise $\mathcal{N}(0, \sigma^2 I)$. The goal is to find $\mathbf{x}$ in the constellation using the above equation. There are various approaches in the literature, but we sketch a linear solution based on lattice reduction. The classical linear solution to this problem is the minimum-mean squared error (MMSE) solution, namely:

$$\hat{\mathbf{x}}_{MMSE} = Q_\Omega \left( (\sigma^2 I + B^T B)^{-1} B^T \mathbf{y} \right),$$

where $Q_\Omega$ quantizes its input to the nearest constellation point. The MMSE linear MIMO detection has in general large performance gap with respect to the Maximum Likelihood (ML) solution. However, it has been shown that the lattice reduced version of $B$ provides a closed performance to the ML solution (Wubben et al., 2004). It is enough to find the unimodular matrix $U$ such that $BU^{-1}$ is the lattice-reduced version of $B$. Then we can plug $BU^{-1}$ to the linear MMSE solution and quantized to the transformed lattice $U\Omega$:

$$\hat{\mathbf{x}}_{LR-MMSE} = Q_\Omega \left( U^{-1} Q_{U\Omega} \left( (\sigma^2 I + U^{-T} B^T B U^{-1})^{-1} U^{-T} B^T \mathbf{y} \right) \right).$$

Not only this improves the performance, measured in terms of bit error rate, but it has been shown that it can achieve asymptotic information theoretic optimality (Taherzadeh et al., 2007). This is the main motivation for using lattice reduction in MIMO setting.

---

[3]In wireless literature, the channel is typically denoted by $\mathbf{H}$. However, to be consistent with our lattice notation, we use $B$.

### D.3 Wireless Simulation

For our simulation, we have used the standard TDLA30 channel model as specified in the 3GPP document TS 38.101-4 Annex B.2.1 with a Doppler shift of 100 Hz. In this model, the wireless channel is modelled by its impulse response. The impulse response of the channel is specified by a fixed number of delays, each corresponding to a communication path, and their respective gain. TDLA30 has 12 distinct paths with maximum delay as 290ns. Each path gain follows Rayleigh distribution with the power gain specified in Table B.2.1.1-2. The correlations are modelled using the Kronecker model. Each correlation profile is specified by a receive and transmit side correlation matrix $R_{rx}$ and $R_{tx}$. The correlated channel is given as

$$R_{rx}^{1/2} B (R_{tx}^{1/2})^T.$$

We use the pre-defined channel profiles for our experiments.

