# OpenReview forum: "Neural Lattice Reduction: A Self-Supervised Geometric Deep Learning Approach"
_TMLR — Accepted by TMLR_

### Review · Reviewer_YZRS · 2024-11-17

**Summary Of Contributions:**

In this paper, the authors consider solving the lattice reduction problem, a combinatorial optimization problem, with deep learning. Namely, given a lattice basis as input, train a neural network to output a base-change uni-modular matrix, where the reduced basis is as close to orthogonal as possible. The crucial feature of the setup is (1) the algorithm does not rely on solved problem solution during the learning process; (2) equivariant and invariant symmetry in the mapping is baked into the architecture which form a better ansatz for neural network parametrization.

The approach involves minimizing the orthogonality defect for $B\varphi_\theta(B)$ where $\varphi_\theta(B)$ is the output of a parametrized NN which is a discrete uni-modular matrix acting on a given basis $B$ of a lattice. In order for the output to be valid,  the $\varphi$ is modeled as a sequence of successive Gauss moves with appropriate rounding. Additionally, three types of invariance/equivariance properties are incorporated by either input transformation or leveraging the GNN architecture and restricting to the hyperoctahedral group.

The result is compared against LLL, a classical algorithm for approximately solving the problem and extension that can jointly solve multiple correlated instances to amortize the computation cost is discussed.

**Audience:**

Yes

**Broader Impact Concerns:**

No concerns involved.

**Claims And Evidence:**

Yes

**Requested Changes:**

If $\varphi$ is the GNN as defined in Section 4.2, and $Q$ is a product of recursive transformations as defined in eqn (6), doesn't backpropagation requires unrolling all the updates and is memory-intensive, at least if the whole pipeline is trained end-to-end? I'd appreciate some clarification.

Given that (1) there are various relaxations made so one can work with optimizing over discrete state space; (2) Proposition 4.2 is only a representation result, and not about whether the algorithm will find the best matrix for objective eqn (5) in such a space, any guarantee on the output given by the proposed algorithm with the chosen parametrization / approximation is desirable. To me, computational complexity such as those given in Section 4.3 is a bit out of context without any quantification on such an error.

Minor comment:

- Section 2: "rises differentiability challenges" -> raises

- Figure 3, I think you are missing a normalization block after the Gram matrix block

- Last sentence before Section 5: you probably mean "the cost per lattice is
potentially reduced"

**Strengths And Weaknesses:**

The authors propose an interesting approach to solve a highly relevant problem in practice. As far as I can tell, the methodology is sound, and there are a few places where novel ideas are introduced for designing the architecture tailored to this specified problem. The paper is also well-written and easy to follow.

In terms of weaknesses, the authors insist on the benefit of not using data. But I imagine there are data simply already existing for any particular application, or in a database collected by mathematicians online. It seems to me rather wasteful to not use these information for e.g., learning better Gauss moves or other places where heuristic choices are made if one has access to some $(B,Q)$ pairs. Even warm-starting from the output of LLL and fine-tuning with NN seems like a good idea. I'm a bit missing on the motivation of not using any data or existing knowledge about the solution at all, and training in a fully agnostic way, which also requires retraining for new problems / hard to generalize across instances.

The performance of the algorithm, as judging from e.g., Table 1, isn't much better than LLL, which seems to suggest either more inductive bias in the architecture is required to aid the optimization, or some form of ``supervised learning" is needed for the NN-based methodology to work (also related to my previous point).

---

> ### Author Response · Authors · 2024-12-16
>
> We thank the reviewer for the detailed and constructive feedback. We wish to address the points raised in the review.
>
> First, we wish to comment on the reviewer's suggestion to **supervise and/or pre-train** our method with the outputs from the LLL algorithm. While we find this option appealing, we believe that it is currently incompatible with our model and training setup.
> The issue is that there is no clear way to decompose the LLL algorithm into a sequence of (extended) Gauss moves, which are the outputs of our model. Therefore, we do not see a straightforward way to leverage supervision and/or pretraining via LLL.
> Nevertheless, we agree with the reviewer that some form of supervision might be useful; we believe exploring different supervision strategies is a promising research direction to improve our model but is beyond the scope of the current work - which, in our view, focuses on the possibility of learning lattice reduction in an unsupervised way.
>
> Regarding the recursive formulation of the model (Equation 6), we agree with the reviewer that directly **back-propagating through the recursion** is memory-intensive for long sequences. In our implementation, we actually circumvented this issue by detaching the output $T_i$ (notation from Equation 6) of the network at each step $i$, and optimizing a cumulated loss (Equation 5) across steps. As a consequence, the model learns to output an optimal Gauss move given the current input lattice, but the whole lattice reduction process is not optimized end-to-end.
> We thank the reviewer for bringing up this point, and we apologize for omitting this implementation detail in the paper. We believe it is important to clarity, and we have added it in Section 4.1 of the re-uploaded version of the manuscript.
>
>
> Next, we wish to comment on the issue raised on **guarantees and error bounds** for the proposed method. We wish to highlight  that the discretization procedure from Section 4.1 can, in principle, output arbitrary unimodular matrices. Since the Gumbel softmax trick enables to represent arbitrary distributions over indices $(i,j)$, the procedure enables to output arbitrary (extended) Gauss moves, which then can produce an arbitrary element of $\text{SL}_n(\mathbb{Z})$ by Proposition 4.2. Now, the GNNs we deploy can approximate arbitrary message-passing GNNs [1], which in turn are universal approximators, i.e., can approximate arbitrary (continuous) functions over the graph nodes [2]. Therefore, our model, if large enough, can approximate the lattice reduction problem - i.e., minimizing the loss from Equation 5 - arbitrary well. It is likely that providing more precise bounds and convergence guarantees is extremely challenging, or even intractable from a theoretical perspective. Neural networks and their learning dynamics are notoriously hard to analyze mathematically, and very little convergence guarantees and error bounds are known. Moreover, such bounds would depend on the specific architecture (a hyperoctahedral-equivariant Graph Neural Network, in our case), the data distribution, and the optimizer. Yet, we agree with the reviewer that this remark on expressivity of GNNs is important, and we have incorporated it in Section 4.2 of the re-uploaded version of the manuscript.
>
>
>
> Lastly, we thank the reviewer for the typos and inaccuracies found. We have corrected them in the re-uploaded version of the manuscript.
>
>
>
> [1] Maron et al., Invariant and equivariant graph networks, 2018.
>
> [2] D'Inverno et al., On the approximation capability of GNNs in node classification/regression tasks, 2023.

---

### Review · Reviewer_sW3q · 2024-12-03

**Summary Of Contributions:**

The paper describes a method for lattice reduction using a differentiable model that includes a learnable graph neural network (GNN. Lattice reduction refers to a mapping from general linear group over reals to general linear group over integers. The neural model is optimized in a self-supervised manner by minimizing the orthogonality defect of basis vectors of a lattice. The paper uses symmetries of lattice reduction to come up with an implementation that works on small-scale problems. The paper empirically analyzes the proposed algorithm with synthetic datasets and on a setup that involves wireless communications.

**Audience:**

Yes

**Broader Impact Concerns:**

No concerns were noted in the paper. However, the reviewer believes this maybe unnecessary although the authors are encouraged to consider including a broader impact statement.

**Claims And Evidence:**

No

**Requested Changes:**

1. Are the poor performance results of the proposed method due to model size limitation, data limitation or something fundamental?

2. Why would knowledge distillation noted in the paper be a good method to simplify computational burden? Especially in light of Question 1 above.

**Strengths And Weaknesses:**

# Strengths

- The paper develops a differentiable approach to solve a problem in combinatorial optimization
- The paper clearly describes how various symmetries involved with lattice reduction can be used to design a differentiable architecture (I am specifically talking about Section 4.2)
- The experiments using wireless communications could be of interest to the applied members of the TMLR community

# Weaknesses

- The empirical results provided in Table 1 suggests that using machine learning may not be the best approach for lattice reduction. Is the poor performance due to the size of the models considered in the paper? Is the bad performance due to a fundamental limitation?

- The proposed method (or algorithm) is highly specialized due to the various assumptions involved in the lattice reduction problem considered in the paper. As a result, I am not convinced about the general message that the TMLR audience might take away after reading the paper. I understand that the paper claims that the proposed method is the first to apply learning to lattice reduction but the learning component seems to be very problem-specific and task-specific.

- The proposed method is not computationally any less burdensome than the baseline LLL approach.

---

> ### Author Response · Authors · 2024-12-16
>
> We thank the reviewer for the constructive feedback and the comments. We wish to address the questions raised in the review.
>
> We acknowledge that our method does not generally out-compete LLL in our experiments.
> We do not believe that this is a fundamental limitation, but rather a problem of scaling up the model and the training, i.e. the network can, in principle, learn better solutions than LLL, at a sufficiently-large scale.
>
> For example, our empirical results in Fig. 5 suggest scaling the number of extended Gauss moves ($k$, or "Steps" in Fig.5) deployed: the plots show that with large enough $k$, our model not only reaches the performances of LLL, but even out-competes it on some lattices (consider the lower end of the shaded region, corresponding to mean minus standard deviation).
> Theoretically, this is motivated by the fact that our model can output arbitrary unimodular matrices, if enough (extended) Gauss moves are allowed (Theorem 4.1 and Proposition 4.2).
>
> We believe that performances can also be improved by training larger models, with the same architecture. Theoretically, the Graph Neural Networks we deploy are universal approximators for functions over graphs -- we added a remark on this in Section 4.2 of the re-uploaded manuscript.
> Moreover, very wide architectures are notoriously beneficial not only for the increased model expressivity, but also for their improved training dynamics.
>
> While larger architectures and $k$ result in higher computational cost, distillation can decrease the computational cost at inference time. For example, it might be possible to distill the trained model into another one with a small architecture that factors the unimodular matrix $Q$ in fewer than $k$ matrices. The latter avoids running the model $k$ times at test time to produce the reduced lattice, amending for the computational cost inherent in training with a large $k$. We acknowledge that this was not clear in Section  4.3 -- we have elaborated on this in the re-uploaded version of the manuscript.
>
> Overall, our main goal was showcasing that a neural model can address lattice reduction, with comparable performances to a popular classical algorithm (LLL). Therefore, in our experiments, we kept compute cost in mind, striving to obtain a model balancing low computational cost and good performance, i.e., comparable to LLL.

---

> > ### Comment · Reviewer_sW3q · 2024-12-27
> >
> > I thank the authors for the rebuttal. I would like the authors and AE to note that I have carefully read the comments made by other reviewers as well as the authors rebuttal. At this point, I do not want any further changes in the paper. however, I'd like to note the following comments in response to rebuttal:
> >
> > > We acknowledge that our method does not generally out-compete LLL in our experiments. We do not believe that this is a fundamental limitation, but rather a problem of scaling up the model and the training, i.e. the network can, in principle, learn better solutions than LLL, at a sufficiently-large scale.
> >
> > I defer to the authors expertise in the topic. I would have liked to see an empirical demonstration of the above result in the paper as that would have significantly strengthened the argument for using neural nets for solving a method that continues to solved algorithmically (not via learning) (For example see https://epubs.siam.org/doi/10.1137/23M1557933)
> >
> > > Overall, our main goal was showcasing that a neural model can address lattice reduction, with comparable performances to a popular classical algorithm (LLL). Therefore, in our experiments, we kept compute cost in mind, striving to obtain a model balancing low computational cost and good performance, i.e., comparable to LLL.
> >
> > My main remaining concern (not major) is that the choice above might perhaps provide an incomplete picture. Also as another reviewer mentioned below, a naive fully-supervised baseline would have been helpful as well as its not clear whether the machinery proposed in the paper is necessary.

---

> > > ### Author Response · Authors · 2025-01-02
> > >
> > > We thank the reviewer for carefully reading our comments. We are glad the reviewer does not have remaining major concerns.
> > >
> > > We agree with the reviewer that our experiments do not provide exhaustive evidence to strongly recommend the use of neural nets over LLL. Indeed, our choice above is not intended to answer this question in a complete way - which involves many factors such as overall accuracy, complexity, memory usage, parallelizability, or hardware support - but rather focuses on a single aspect, i.e.,  the learnability of this task.
> > >
> > > Regarding the fully-supervised baseline, we agree that (some form of) supervised training is appealing, but we do not see this strategy as simpler than ours. Indeed, most of the machinery proposed in our work would still be necessary in a fully-supervised context, e.g., dealing with the symmetries, outputting invertible integer matrices, and optimizing over them. Moreover, a supervised approach would additionally require outputting more flexible matrices to match with the ground-truth ones, and defining a new suitable loss. Instead, our self-supervised approach enables us to simply output (extended) Gauss moves, and to optimize the orthogonality defect directly. Hence, we believe the supervised strategy can be a promising extension of our method worth further exploration, rather than a simpler baseline.

---

### Review · Reviewer_457A · 2024-12-08

**Summary Of Contributions:**

The authors consider the application of a self-supervised deep neural network to the problem of lattice reduction, where the network has bespoke invariance/equivariance modes built-in that suit the lattice reduction problem.  More specifically, they propose a novel deep learning model which is invariant and equivariant to various discrete groups for lattice reduction outputting unimodular matrices via a self-supervised objective. They compare their method to the state-of-the-art method the Lenstra–Lenstra–Lovász method, a non-ML method. Finally they show that they show that their network may be trained simultaneously on on several lattices in parallel which may yield a good performance/complexity tradeoff if the lattices are correlated.

**Audience:**

Yes

**Claims And Evidence:**

Yes

**Requested Changes:**

I think that the work would be strengthened with a discussion of difficulties working over $GL_n(\mathbb Z)$ and, in particular what the 'gap' is between full right modular equivariance and hyperoctahedral equivariance. What are some elements of the $GL_n(\mathbb Z)$ that are 'far away' from elements of $H_n$? What properties of $GL_n(\mathbb Z)$ (specifically) make it subtle or hard to design around? Indeed it does seem to be a subtle group, but some exploration of its subtleties and why they lead to difficulties would be a valuable addition.

**Strengths And Weaknesses:**

Strengths
- The paper is extremely well-written an organized. The presentation is well-reasoned, well-organized, and cleanly written.
- The paper features a good balance of new pure theoretical developments with Prop 4.2 and the developments of section 4.2.
- The numerical results give support to the use of self-supervised learning to solve this combinatorial problem. In some settings, the self-supervised model is competitive with the LLL method.
- I quite like how the authors designed a geometric network that is invariant/equivariant w.r.t. the algebraically interesting group $GL_n(\mathbb Z)$. There is a lot of geometric ML work on groups that are less algebraically subtle (e.g. $SO_n(\mathbb R)$), and this reviewer appreciates that the authors were up to the challenge.

Weaknesses
- The benefits of using the proposed self-supervised method over the state-of-the-art are nebulous. The actual performance of the trained network is at best 'in the same ball park' as the state of the art method and, at worse, considerably worse (Table 1).
- The authors ultimately fail to construct a network that is equivariant/invariant w.r.t. the full group of interest (Right unimodular equivariance). This is a minor criticism, however, given the algebraic subtlety of $GL_n(\mathbb Z)$, as discussed in the strengths section.

---

> ### Author Response · Authors · 2024-12-16
>
> We thank the reviewer for the comments and the constructive feedback.
>
> We appreciate the suggestion of discussing more thoroughly the challenges around $\text{GL}_n(\mathbb{Z})$-equivariance, and we wish to comment on this. As mentioned in the review, the unimodular group $\text{GL}_n(\mathbb{Z})$ is notoriously mysterious, and its structure is not understood. Intuitively, this is due to the arithmetic nature of the modular group; understanding it involves subtle questions in number theory. Therefore, constructing $\text{GL}_n(\mathbb{Z})$-equivariant maps - even linear ones - is extremely challenging. To exemplify this, the standard group-theoretical technique of constructing linear equivariant maps by leveraging Schur's Lemma [1] requires knowing the (irreducible) representations of the given group, and applies only to compact groups. The representation theory  of $\text{GL}_n(\mathbb{Z})$ is extremely involved [2], and the group is not compact.
> The challenge of constructing equivariant maps is also partially discussed in Appendix B.2, where we analyze $\text{GL}_n(\mathbb{Z})$-equivariant maps $\mathbb{R}^{n \times n} \rightarrow \mathbb{R}^{n \times n}$ (with the unimodular group acting via matrix multiplication), and in particular show that there are no non-trivial such maps that are multi-linear.
>
> We believe that the hyperoctahedral group $\text{H}_n$ provides a good compromise for designing an equivariant network. First, this group and its desired equivariance property (Equation 8 in our paper) are significantly simpler, allowing to deploy certain kinds of Graph Neural Networks (GNNs), as explained at the end of Section 4.2. Moreover, the hyperoctahedral group contains all the integral orthogonal matrices, i.e., $\text{H}_n = \text{GL}_n(\mathbb{Z}) \cap \text{O}_n(\mathbb{R})$. Due to singular value decomposition, the matrices in $\text{GL}_n(\mathbb{R})$ are obtained from the ones in $\text{O}_n(\mathbb{R})$ by rescaling basis vectors in a given base. Therefore, equivariance to hyperoctahedral/orthogonal group is not far from equivariance to the full general group; the missing symmetries correspond to anisotropic rescalings. Even further, the (isotropic) scaling symmetry of our model (third bullet point on page 7) partially amends to this.
>
>
> We agree with the reviewer that the above discussions are important, and we have incorporated them in Section 4.2 of the re-uploaded version of the manuscript.
>
> Regarding the benefits of our method over state-of-the-art, we want to emphasize the goal of this work is exploring the possibility of learning a lattice reduction algorithm.
> Moreover, as argued in our comments to the other reviewers, the performance gap between our method and LLL is *relatively* small and can be likely further reduced by scaling (see the relative performance gap in Table 1).
>
>
> [1] Behboodi et al., A PAC-Bayesian generalization bound for equivariant networks, 2022.
>
> [2] Putman, The representation theory of $\text{SL}_n(\mathbb{Z})$.

---

### Author Response · Authors · 2024-12-16

We thank all the reviewers for their comments and the helpful feedback.
We hope our replies below help clarifying their doubts and answer their concerns.

We have also uploaded a revised manuscript which incorporates their feedback and includes a few additional experiments.
We *highlighted in red* all the changes with respect to the previous version.
In summary, we included:

- A paragraph about the training strategy adopted to avoid back-propagating through the recursive steps.
- A paragraph describing the gap between the chosen equivariance group $H_n$ and the full symmetry group $\text{GL}_n(\mathbb{Z})$.
- Additional comments on the possibility of using distillation to reduce the cost of our method.
- Additional results in Table 1, including an additional MLP baseline to highlight the importance of adopting an equivariant design, and the relative performance difference (*gap*) between LLL and our method to highlight the small gap.
- Other corrections suggested by the reviewers.

---

### Decision · Action_Editor_CzyU · 2025-01-03

**Recommendation:** Accept as is

**Comment:**

All three reviewers are generally satisfied after the discussion process and recommend acceptance. They do note that the performance of the proposed method is not very high and that results with a supervised neural network baseline/upper bound is missing.

**Audience:**

The paper presents an unsupervised neural network-basedmethod for lattice reduction. Readers interested in combinatorial optimization would be interested in such a paper.

**Claims And Evidence:**

Yes, all claims are supported.